# Non-Invasive Evaluation of Retinal Vascular Alterations in a Mouse Model of Optic Neuritis Using Laser Speckle Flowgraphy and Optical Coherence Tomography Angiography

**DOI:** 10.3390/cells12232685

**Published:** 2023-11-22

**Authors:** Seth E. Buscho, Fan Xia, Shuizhen Shi, Jonathan L. Lin, Bartosz Szczesny, Wenbo Zhang, Massoud Motamedi, Hua Liu

**Affiliations:** 1Department of Ophthalmology and Visual Sciences, University of Texas Medical Branch, Galveston, TX 77555, USA; sebuscho@utmb.edu (S.E.B.); faxia@utmb.edu (F.X.); shushi@utmb.edu (S.S.); jolin@utmb.edu (J.L.L.); baszczes@utmb.edu (B.S.); we2zhang@utmb.edu (W.Z.); mmotamed@utmb.edu (M.M.); 2Department of Pharmacology and Toxicology, University of Texas Medical Branch, Galveston, TX 77555, USA; 3Department of Anesthesiology, University of Texas Medical Branch, Galveston, TX 77555, USA; 4Department of Neurobiology, University of Texas Medical Branch, Galveston, TX 77555, USA

**Keywords:** multiple sclerosis (MS), optic neuritis, laser speckle flowgraphy (LSFG), optical coherence tomography angiography (OCTA), retinal vasculature, retinal blood flow, retinal vascular inflammation, biomarker

## Abstract

Optic neuritis, a characteristic feature of multiple sclerosis (MS), involves the inflammation of the optic nerve and the degeneration of retinal ganglion cells (RGCs). Although previous studies suggest that retinal blood flow alterations occur during optic neuritis, the precise location, the degree of impairment, and the underlying mechanisms remain unclear. In this study, we utilized two emerging non-invasive imaging techniques, laser speckle flowgraphy (LSFG) and optical coherence tomography angiography (OCTA), to investigate retinal vascular changes in a mouse model of MS, known as experimental autoimmune encephalomyelitis (EAE). We associated these changes with leukostasis, RGC injury, and the overall progression of EAE. LSFG imaging revealed a progressive reduction in retinal blood flow velocity and increased vascular resistance near the optic nerve head in the EAE model, indicating impaired ocular blood flow. OCTA imaging demonstrated significant decreases in vessel density, number of junctions, and total vessel length in the intermediate and deep capillary plexus of the EAE mice. Furthermore, our analysis of leukostasis revealed a significant increase in adherent leukocytes in the retinal vasculature of the EAE mice, suggesting the occurrence of vascular inflammation in the early development of EAE pathology. The abovechanges preceded or were accompanied by the characteristic hallmarks of optic neuritis, such as RGC loss and reduced visual acuity. Overall, our study sheds light on the intricate relationship between retinal vascular alterations and the progression of optic neuritis as well as MS clinical score. It also highlights the potential for the development of image-based biomarkers for the diagnosis and monitoring of optic neuritis as well as MS, particularly in response to emerging treatments.

## 1. Introduction

Multiple sclerosis (MS) is a progressive autoimmune disorder marked by the demyelination of the central nervous system (CNS). It primarily presents in young adulthood and is estimated to affect 2.8 million people worldwide [1]. MS severely affects the quality of life of patients by inhibiting their ability to work, socialize, pursue recreational activities, and maintain self-care [2]. Symptoms of MS observed in a clinical setting encompass a range of issues, such as impaired vision, movement disorders, dizziness, cognitive dysfunction, slurred speech, and difficulties related to sexual, bowel, and bladder function [1]. Among them, optic neuritis (ON) is a major clinical feature and can lead to visual impairment or even blindness. Around 30–70% of MS patients develop optic neuritis during their disease course, and up to 94–99% of patients with post-mortem examinations show demyelinating plaques in the optic nerves [2].

ON is characterized by inflammation of the optic nerve [3]. It was previously thought that immune cells, particularly activated T cells, were responsible for attacking the myelin sheath surrounding the optic nerve, leading to the demyelination, dysfunction, and degeneration of the optic nerve [3]. The thinning of the retinal nerve fiber layer (RNFL) and the dysfunction and degeneration of retinal ganglion cells (RGCs) occur as a result of the degeneration of the optic nerve, ultimately leading to an irreversible loss of vision [4,5]. Moreover, demyelination-independent neurodegeneration also contributes to the pathology of ON during MS [6]. Beyond RGC degeneration, clinical studies using optical coherence tomography angiography (OCTA) suggest that retinal vessel density and blood perfusion are altered in ON. However, the data are often contradictory. Some studies indicated that eyes from MS patients had significantly decreased retinal vessel density in the superficial vascular complex (SVC) and deep vascular complex (DVC) compared with healthy controls, and that vessel density reduction was even greater in eyes with ON [7,8,9,10,11], associated with reduced oxygen extraction and retinal blood flow [12]; however, other studies did not find any vessel density reduction in MS eyes compared with healthy eyes or a difference between MS eyes with vs. without ON [8,9,10]. There are also many inconsistencies regarding the location of and mechanisms underlying these vascular changes [8,9,10]. Considering the close interaction between retinal vessels and neurons, and that vascular dysfunction may contribute to neurodegeneration, it is important to further clarify whether vascular alterations occur in ON or not.

In addition to the use of OCTA for the non-invasive monitoring and quantification of MS-induced alterations in retinal vascular features, laser speckle flowgraphy (LSFG) has emerged as a valuable non-invasive tool to monitor changes in retinal blood flow under various pathological conditions [13,14,15,16,17,18]. By utilizing the laser speckle phenomenon, LSFG enables the quantitative estimation of blood flow in ocular tissues, including the retina. This technology provides a convenient and highly sensitive method for assessing alterations in retinal blood flow with promising capabilities to capture dynamic changes in blood flow, yielding improvement in the current understanding of the pathophysiology of retinal vascular disorders, thus offering valuable insights for diagnosis, monitoring, and potential treatment strategies for conditions such as MS.

The current understanding of retinal vascular changes in animal models accompanying ON is limited. To address this, we conducted a longitudinal evaluation of retinal vascular alterations in the mouse model of experimental autoimmune encephalomyelitis (EAE), a well-established model for studying ON [19]. Utilizing two clinically relevant non-invasive imaging techniques, OCTA and LSFG, we characterized retinal vascular changes in the EAE model during disease progression. Furthermore, we performed a retinal leukostasis study to characterize the pathophysiology of ON involving inflammation and vascular dysfunction, which allowed us to relate these changes to the findings from non-invasive assessments of retinal vascular alterations in the EAE model using OCTA and LSFG imaging techniques. This study contributes to a deeper understanding of the retinal vascular dynamics associated with ON and has the potential to impact the development of more effective diagnosis and treatment strategies for ON as well as MS.

## 2. Materials and Methods

### 2.1. Animals

In our current pre-clinical study, which received approval from the Institutional Animal Care and Use Committee at the University of Texas Medical Branch, we utilized C57BL/6J wild type (WT) mice (Stock No: 000664) and C57BL/6-EGFP (Stock No: 006567) from the Jackson Laboratory (Bar Harbor, ME, USA). The mice were bred in the animal care facility at the University of Texas Medical Branch. All of the experimental procedures and animal usage strictly adhered to the guidelines provided by the Association for Research in Vision and Ophthalmology Statement for the Use of Animals in Ophthalmic and Vision Research.

### 2.2. EAE Model

We chose to exclusively use female mice for this study, because C57BL/6J males do not display severe clinical symptoms and reach the maximum clinical score [20]. At 13 weeks of age, we induced the EAE model using the MOG 35-55/CFA EMULSION PTX kit (#EK2110, Hooke Laboratories, Lawrence, MA, USA), as previously described [4]. Briefly, an emulsion containing myelin oligodendrocyte glycoprotein (MOG) 35-55 peptide and complete Freund’s adjuvant was subcutaneously injected into each hind leg (0.1 mL/leg). At 2 and 24 h after injection, pertussis toxin (PTX) (100 ng/mouse) was administered intraperitoneally. The control mice were injected with an emulsion containing complete Freund’s adjuvant without MOG 35-55 peptide and PTX. After the induction of immunization, optic neuritis typically starts at 2 weeks and peaks at 3 weeks [5]. The clinical score for EAE (*n* = 7) and control (*n* = 5) mice was obtained at 0, 7, 10, 14, 21, and 28 days post-immunization (dpi), in accordance with the manual of the MOG 35-55/CFA EMULSION PTX kit and based on changes in the motor functions of the tail, legs, and neck. Rapid increases in the clinical score usually occur between 12 and 18 dpi [21].

### 2.3. Bone Marrow (BM) Transplantation and Scanning Laser Ophthalmoscope (SLO)

BM transplantation was performed as previously described [22,23]. Briefly, 8-week-old WT mice were irradiated at a dose of 850 rads (8.5 Gy) with a Gammacell 40 irradiator (MDS Nordion, Ottawa, ON, Canada). A lead shield was placed on the head and eyes of each mouse to protect the head and eyes from potential radiation injury. We subsequently obtained BM cells from the femurs and tibias of donor GFP mice, followed by washing, counting, and resuspension in phosphate-buffered saline (PBS). Within 24 h of irradiation, we injected 200 µL of the cell suspension (1.0 × 10^7^ cells) into WT recipient mice via tail vein. After 4 weeks, the EAE model was induced in WT recipient mice. Right before or 7 days after EAE induction, SLO imaging was performed. Briefly, the mice were injected intraperitoneally with 100 mg/kg ketamine and 10 mg/kg xylazine for anesthesia, and their eyes were dilated with topical tropicamide and phenylephrine. The mice were then placed on a heated platform, and the Heidelberg Spectralis HRA system (Heidelberg Engineering, Franklin, MA, USA) was used to record in vivo movement of GFP-tagged leukocytes in the retinas. The frame rate was set to 5 frames/second with a 55° field of view, and each image consisted of 100 frames. Leukocytes were counted for each retina (*n* = 6 pre-EAE, *n* = 6 EAE).

### 2.4. Leukostasis

Retinal leukostasis was performed as previously described [22,24,25]. In short, 14 days after EAE induction, the anesthesia of the mice was achieved with an intraperitoneal injection, consisting of 100 mg/kg ketamine and 10 mg/kg xylazine. The chest cavity was opened, and the right atrium was incised to allow for outflow. A perfusion catheter was inserted into the left ventricle. Following the removal of erythrocytes and nonadherent leukocytes using PBS, adherent leukocytes and retinal vasculature were labeled by perfusion with rhodamine-coupled concanavalin A (Con A) lectin (40 g/mL in PBS, pH 7.4; Vector Laboratories, Burlingame, CA, USA). PBS was subsequently re-perfused to eliminate any remaining unbound Con A. The eyeballs of the mice were collected and fixed overnight with 4% paraformaldehyde (PFA) at 4 °C, and then their retinas (*n* = 6 Control, *n* = 6 EAE) were dissected for the quantification of the total number of adherent leukocytes.

### 2.5. LSFG Imaging

The retinas of the EAE and control mice were imaged using an LSFG-Micro System (Softcare Co., Ltd., Fukuoka, Japan) at 7, 10, 14, and 28 dpi (*n* = 9–10 Control, *n* = 14–16 EAE). Prior to LSFG imaging, the anesthesia of the mice was achieved with an intraperitoneal injection, consisting of 100 mg/kg ketamine and 10 mg/kg xylazine. The eyes were dilated with one drop of phenylephrine and one drop of tropicamide. GenTeal^®^ Tears Eye Gel (Alcon Laboratories, Inc., Fort Worth, TX, USA) was lightly applied to the mouse corneas, a circular glass slide was placed on top of the gel, and the mice were then positioned on the LSFG imaging platform. Images were acquired with the optic nerve head (ONH) in the center over a 4 s (118 frames) period at a rate of 30 frames per second to produce a 3.2 × 2.5 mm^2^ composite area. At each timepoint, each eye was measured three consecutive times without changing positions between each scan. The average of these values was used for subsequent statistical analyses.

### 2.6. LSFG Analysis

LSFG images were analyzed using the LSFG Analyzer software (version 3.00) (Softcare Co., Ltd., Fukuoka, Japan) (Figure 1). Recorded parameters are described in Table 1. First, the mean blur rate (MBR) of the vessel area (MV), tissue area (MT), and overall area (MA) was quantified within a 200-pixel diameter circle centered around the ONH. Second, waveform analysis was performed, which yielded the beat strength (BS), a marker of the fluctuation in blood flow at the onset of a heartbeat; the BS over MBR (BOM), a marker of peripheral vascular resistance; the maximum MBR over a cardiac cycle (Max MBR); the minimum MBR over a cardiac cycle (Min MBR); and the Max–Min MBR over a cardiac cycle. Third, the volume of blood flow through retinal arteries and veins was quantified as the total retinal flow index divided by the vessel width (TRFI/w), using the total retinal artery and vein analyzer (TRAVA). All of the examiners were blinded to the mouse treatment during LSFG image acquisition and analysis.

### 2.7. OCTA Imaging

OCTA scans were performed for each retina (*n* = 14 Control, *n* = 16 EAE) at 14 dpi. Prior to imaging, the anesthesia of the mice was achieved with 3.8% isoflurane using a Digital Low-Flow Anesthesia System (Kent Scientific, Torrington, CT, USA), and phenylephrine and tropicamide were administered to their eyes to induce mydriasis. Next, the mice were wrapped in cotton gauze to inhibit their movement and maintain their body temperature. They were subsequently placed on a heating pad on the imaging platform with a continuous supply of isoflurane. GenTeal^®^ Tears Eye Gel was applied to the eye that was not actively being imaged to prevent corneal clouding. OCTA images were acquired using the Heidelberg HRA + OCT Spectralis (Heidelberg Engineering, Heidelberg, Germany), as previously described [26]. Briefly, a 20° by 20° volume was captured of each quadrant of the retina using the 30° lens and mouse adaptor lens.

### 2.8. OCTA Analysis

OCTA scans with significant image artifacts or a quality-control index rated at 25 dB or less, as determined by the Heidelberg Eye Explorer version 6.12.4.0, were omitted from data analysis. The vessel density, total vessel length, and total number of junctions were recorded for each OCTA image using the AngioTool software (National Cancer Institute, Bethesda, MD, USA) [26,27]. The mean value was subsequently calculated across all imaged quadrants for one retina to yield each data point for statistical analysis. The examiners were blinded to the mouse treatments during OCTA image acquisition and analysis.

### 2.9. Visual Acuity

The visual acuity of the mice (*n* = 9 Control, *n* = 14, EAE) was measured based on an innate visual–motor reflex using a plexiglass testing chamber and the software OptoMotry [28]. In brief, the mice were placed on a platform, which was surrounded by four identical screens displaying changeable moving grids. The mice were able to move freely with a limited range on the platform, and an experimenter tracked the mouse’s head movement with a crosshair. The animal was determined to be able to visualize the stimulus if its head tracked the stimulus resulting from the cylinder’s rotation relative to the stationary arms of the crosshair.

### 2.10. Retinal Flatmounts

At 28 dpi, the mouse eyeballs were fixed in 4% PFA at 4 °C overnight. Following dissection, the retinas were washed with PBS, blocked, and then permeabilized, using PBS that contained 5% donkey serum and 0.3% Triton-X-100 (ThermoFisher Scientific, Waltham, MA, USA), for a period of 3 h. Next, they were incubated with an antibody against RBPMS (1:200; MilliporeSigma, Burlington, MA, USA), a marker for RGCs, at 4 °C overnight. The next day, the retinas were incubated with an Alexa Fluor 594-conjugated secondary antibody (1:400; ThermoFisher Scientific, Waltham, MA, USA) at 4 °C for 4 h. In the end, the retinas (*n* = 9 Control, *n* = 14 EAE) were mounted, and images were captured using confocal microscopy (LSM 800, Carl Zeiss Inc., Thornwood, NY, USA).

### 2.11. Statistical Analysis

The mean and the standard error of the mean (SEM) for both groups were reported. To compare the EAE and control mice, the student’s *t*-test was undertaken using the GraphPad Prism program (GraphPad Software Inc., La Jolla, CA, USA). The alpha value was designated at 0.05, such that a *p* < 0.05 was considered statistically different.

## 3. Results

### 3.1. Establishment of the EAE Model

First, we sought to establish the EAE model in the mice. After disease induction, the mice were monitored for changes in body weight and EAE clinical scores, which were evaluated based on the appearance of clinical signs of EAE at 0, 7, 10, 14, 21, and 28 dpi (Figure 2A,B). While there were no significant differences in body weight and EAE clinical scores for the first 10 days, the body weight and EAE clinical scores of the EAE mice were significantly different from the control mice at 14 dpi and became progressively more severe thereafter. At 28 dpi, the EAE mice exhibited marked weight loss and severely restricted movement, as demonstrated by their high EAE score. Moreover, the staining of retinal flatmounts for RBPMS, which is an RGC marker, demonstrated a significant loss of RGCs in the EAE mice compared to the control mice (Figure 2C). These data indicate the successful establishment of the EAE model.

### 3.2. Characterization of Retinal Vasculature in EAE Using OCTA

OCTA is a dye-free, non-invasive, and cost-effective imaging modality that generates high resolution images of the retinal vasculature. Currently, it is widely used in clinical settings to examine vascular alterations in retinopathies. OCTA distinguishes vasculature from background tissue based on the movement of red blood cells, thus only detecting vasculature that is both structurally present and functionally intact [29,30]. At 14 dpi, we obtained images of the retinal vasculature of the control vs. EAE mice with OCTA and then analyzed the vessel density, counted the number of junctions, and measured the total vessel length within the superficial vascular complex (SVC), intermediate capillary plexus (ICP), and deep capillary plexus (DCP), respectively (Figure 3). Although there was no statistically significant difference in the vascular parameters in the SVC, the EAE mice displayed significantly decreased vessel density, number of junctions, and total vessel length in the ICP and DCP. These data indicate that retinal vascular alterations can be detected non-invasively using OCTA in the EAE model.

### 3.3. Retinal Blood Flow Is Impaired in the EAE Mice

To further characterize retinal vascular alterations in EAE, we utilized the LSFG-Micro System, another non-invasive and dye-free imaging technology, to quantitatively assess ocular blood flow near the optic nerve head (Figure 4). The main output parameter of LSFG is the mean blur rate (MBR), which is calculated based on the rate of speckle pattern blurring resulting from moving blood cells, providing a quantitative index for relative blood flow velocity. In our study, the MBR was calculated over a 4 s (118 frames) period for the vascular area (MV), tissue area (MT), and all of the areas (MA) (Figure 5A). The MV was not greatly affected at 7 and 10 dpi; however, the MT and the MA were significantly reduced in the EAE mice as early as 7 dpi or 10 dpi, respectively. With the progression of EAE, the MV, MT, and MA continued to decline at 14 and 28 dpi.

In addition to the MBR, waveform analysis was also conducted to assess the BS, BOM, Max MBR, Min MBR, and the Max–Min MBR, as described in Table 1. The BS and BOM were found to not be altered early during EAE progression; however, they were significantly increased in the EAE mice at 28 dpi, indicating increased retinal vascular resistance (Figure 5B). The Max MBR and Min MBR were altered beginning at 7 dpi or 10 dpi, but the Max–Min MBR demonstrated no difference at any time point between the control and EAE mice (Figure 5B).

Finally, the total retinal flow index divided by vessel width (TRFI/w) was calculated as the retinal blood flow volume in the top four highest velocity arteries (Figure 1E and Figure 5C). A trend of decrease in TRFI/w was observed in the EAE mice at 14 dpi, and a significant reduction was found at 28 dpi, indicating that peak blood flow volume in large retinal arteries dropped during EAE progression. In sum, our data demonstrated that retinal blood flow velocity is significantly and progressively reduced, while vascular resistance is increased during EAE progression.

### 3.4. Visual Acuity Is Decreased in the EAE Mice

Visual acuity assessment is one method used to monitor disease progression and assess functional impairment in MS patients [31,32,33]. To understand retinal neuronal changes relative to vascular dysfunction, we assessed visual acuity at different time points in the EAE model. We found that visual acuity was significantly decreased in the EAE mice starting at 10 dpi with progressive decline at 14 and 28 dpi (Figure 6). These results support the notion that visual function is impaired in EAE prior to the onset of systemic changes such as weight loss and EAE clinical score, which were observed around at 14 dpi. However, visual acuity was not impaired before the earliest detectable vascular changes, which could be identified with LSFG at 7 dpi.

### 3.5. Retinal Vascular Inflammation Is Increased in EAE

To understand the potential mechanisms underlying reduced blood flow in EAE, we focused on vascular inflammation, as extensive leukocyte recruitment with subsequent adhesion to the vascular endothelium could block blood flow [34,35]. Additionally, cytokines produced by leukocytes or local cells could induce vascular dysfunction, resulting in impaired vessel dilation and increased vessel constriction. Thus, we generated BM chimeric mice, which had leukocytes labeled with GFP, in order to examine leukocyte actions in the retinal vasculature via high resolution SLO imaging. Given that each image was composed of 100 frames, only fixed elements and features within the retina, such as retinal vessels and firmly attached leukocytes, were documented while capturing images. We observed a minimal number of leukocytes adherent to retinal vasculature prior to the onset of EAE, indicating there was no overt retinal vascular inflammation in pre-EAE retina. However, at 7 dpi, the number of attached leukocytes was significantly increased, particularly in retinal veins and capillaries (Figure 7A,B). The marked increase in leukostasis is consistent with a previous report that, in the optic nerve, an upregulation of inflammatory cytokines occurs as early as 7 dpi [36]. Moreover, we assessed leukocyte attachment to retinal vessels using a leukostasis assay at 14 dpi. We perfusion-labeled retinal vasculature and leukocytes with rhodamine-coupled Con A and visualized leukocyte adherence via confocal microscopy. We found that, at 14 dpi, the number of adherent leukocytes in the central and peripheral retinal vasculature was also significantly greater in the EAE mice (Figure 7C–E). Indeed, the EAE mice had an almost 6-fold higher number of adherent leukocytes compared to the control mice. In sum, using two different approaches, we demonstrated that vascular inflammation occurred in the retina early during EAE pathology development.

## 4. Discussion

Using non-invasive imaging modality, LSFG, we provide evidence that retinal blood flow velocity is significantly reduced in a mouse EAE model, which is frequently used to study the mechanisms of optic neuritis. We found that profoundly impaired vessel perfusion occurs at very early stages in EAE as demonstrated by decreased MBR in the tissue area. Initially, blood perfusion within small retinal vasculature progressively decreased, followed by alterations in large retinal vessels accompanying changes in the clinical score and body weight as indicators of disease progression. While there are no other studies that have used LSFG to characterize retinal blood flow velocity in EAE, our results are in line with animal studies that demonstrate hypoxia and hypoperfusion of the inflamed spinal cord occur during EAE [37,38,39]. Consistent with the reduction in retinal blood flow, an analysis of retinal vasculature with OCTA, which constructs high resolution images of retinal vascular structure based on the signal difference generated by erythrocyte motion [29,30], revealed that there was a significant loss of vessel density, total vessel length, and the total number of junctions in the ICP and DCP, two sublayers of the DVC, in the EAE mice. Overall, our results are congruent with several clinical studies that found that decreased retinal blood flow velocity using retinal function imager [40] and decreased retinal vascular density in OCTA scans are present in patients with optic neuritis [40,41,42,43,44,45]. Our findings in conjunction with the study by Bostan et al. also suggest that the DVC may be more sensitive to MS-induced vascular alterations compared to the SVC [45].

The mechanism underlying this pathognomonic decrease in blood flow velocity has yet to be understood, but it is widely known that inflammation could cause reduced blood flow, vascular blockage, and vasoconstriction [34]. We observed a dramatic increase in the number of adherent leukocytes to retinal vasculature in the EAE mice. Hence, it is likely that the decreased MBR we observed with LSFG could be due to an inhibition or blockage of blood flow by leukocytes. Yet, this mechanism does not explain the decreased MV at later stages in EAE, as leukocyte adhesion and extravasation are largely confined to the level of the post-capillary venule, which is not accounted for by MV measurement [46]. Additionally, our results indicate that the BS, a value proportional to the peripheral vascular resistance, increases at a later stage of EAE, which further supports the role of an additional modulator of retinal blood flow velocity. Therefore, it is plausible that decreased blood flow velocity could be the result of vasoconstricting factors produced during vascular inflammation. For example, immune cells such as macrophages, dendritic cells, and T cells could synthesize endothelin-1 (ET-1), which is a potent vasoconstricting factor [47]. Previous studies suggest that ET-1 may play a role in promoting hypoperfusion in the retina, extraocular arteries, and brain of patients with MS [48,49,50]. Additionally, the administration of ET-1 receptor antagonists has shown promise in mitigating the progression of EAE and MS and reversing cerebral hypoperfusion [49,51]. Proinflammatory cytokines such as TNF-α and IL-1β can also impair vasodilation and enhance response to vasoconstrictors [52]. Future studies are needed to elucidate the specific role of leukocytes and leukocyte-derived cytokines in the regulation of retinal blood flow in optic neuritis.

Our results also suggest that vascular impairment occurs prior to RGC dysfunction, as evidenced by a significant decrease in MT beginning at 7 dpi vs. visual acuity decline beginning at 10 dpi. It is still unclear whether vascular alterations could precipitate or contribute to retinal neurodegeneration in optic neuritis in MS. Nonetheless, significant research points towards vascular hypoperfusion playing a role in neuronal compromise through a hypoxia-induced mechanism. A decreased availability of oxygen leads to the decreased mitochondrial consumption of oxygen, with a subsequent depletion of ATP and loss of ATP-dependent metabolic processes [53,54]. One of the most significant ATP-consuming processes in the CNS, the Na^+^/K^+^-ATPase, promotes the extrusion of Na^+^ into the extracellular space, following electrical stimulation under physiologic circumstances. In cases of ATP depletion, the excessive accumulation of intracellular sodium from a loss of the Na^+^/K^+^-ATPase promotes sodium–calcium exchanger activity, resulting in calcium excitotoxicity with subsequent cell death [55]. This has been further explored by studies that used the anti-epileptic sodium channel blockers phenytoin [56], lamotrigine [57], and carbamazepine [58] to treat EAE. One large clinical trial indeed showed that patients treated with phenytoin suffered from 30% less thinning of the RNFL compared to patients treated with a placebo and that phenytoin was neuroprotective against optic neuritis [59]. Another study noted that hypoxic pre-conditioning in EAE resulted in attenuated leukocyte infiltration and a less severe clinical course [60]. These studies, as well as our own, warrant a further exploration of the notion that vascular hypoperfusion is part of the pathogenesis of optic neuritis rather than an epiphenomenon or independent event.

As the number one cause of non-traumatic disabling disease in young adults [61], it is imperative that biomarkers of MS, which can reliably predict the onset and progression of the disease, are identified. At present, the gold standard methods of diagnosing MS, including magnetic resonance imaging (MRI) and cerebrospinal fluid (CSF) analyses, are not suitable for a large-scale screening of populations at risk. MRI is expensive, while CSF analysis is invasive by nature, thus increasing a patient’s risk of complications [62,63]. Most of the previously identified biomarkers are either invasive or are inadequate in their reliability, limiting their applicability to clinical settings [64]. As an extension of the brain, the retina offers an ideal platform to diagnose and monitor pathologic changes within the CNS, including those caused by MS, through non-invasive imaging. A quantitative monitoring of the thinning of the RNFL or the composition of the ganglion cell layer and inner plexiform layer using OCT has emerged as a valuable tool for assessing MS-induced structural changes in the retina [65], offering potential for diagnosing and monitoring MS progression. While conventional OCT imaging provides high resolution images of the retinal structure, it is not suitable for assessing MS-induced vascular changes in the retina. However, recent advancements in OCTA technology have allowed for the evaluation of MS-induced retinal vascular alterations, providing a functional imaging modality for monitoring and quantifying these changes [10]. This functional imaging approach, combined with laser speckle analysis enabling the visualization and quantification of vascular dysfunction as demonstrated in this current study, offers a comprehensive assessment of both morphological and functional aspects of the retina, enhancing our ability to detect early signs of MS. Certainly, there is a limitation in using LSFG and OCTA, since optic neuropathies are caused by various disorders—not only MS but also trauma, sinusitis, and so on. Integrity of imaging approach with other biomarkers and patient’s history is needed to rule out other possibilities and increase its specificity to diagnose optic neuritis associated with MS.

In summary, we provide original evidence that the retina undergoes significant hypoperfusion during the course of optic neuritis in a mouse EAE model. These pathologic alterations in the retina of the EAE mice clears contradictions in clinical studies and supports clinical studies showing decreased retinal vessel density in optic neuritis, similar to vascular changes observed in the brain and spinal cord of MS patients [45,66,67,68]. Moreover, we were able to precisely demonstrate that retinal vascular alterations predate the onset of impairment in both the visual function and systemic signs of EAE. As such, the identification of vascular changes with noninvasive imaging offers a unique opportunity for the early initiation of therapeutic intervention and potentially superior patient outcomes. This study is the first to study retinal vascular changes in an animal ON model. As animal models have a much higher degree of reproducibility than clinical studies, it would be advantageous to further explore retinal vascular alterations in an EAE model to unravel its underlying mechanisms and determine the cause-and-effect relationship between vessel dysfunction and neurodegeneration in optic neuritis. Our data using quantitative OCTA and LSFG, two cost-effective imaging modalities suitable for rapid patient screening based on the visualization and quantification of retinal vasculature alterations, have shown promise in assessing MS-induced vascular changes in the retina of a suitable small animal model for MS, potentially yielding image-based biomarkers that can serve as valuable tools for diagnosing and monitoring MS progression. Continued research in this direction holds promise for improving the accuracy and efficiency of MS diagnosis, ultimately leading to better outcomes for patients.

## Figures and Tables

**Figure 1 cells-12-02685-f001:**
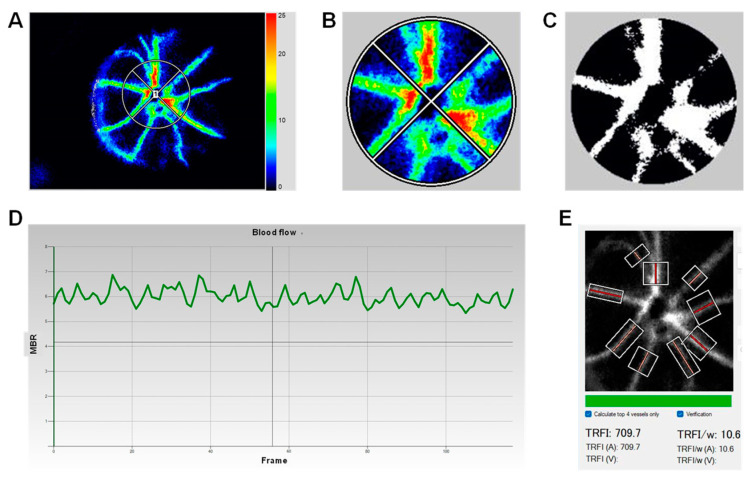
Illustration of the analysis of MBR and waveform values from LSFG. (**A**) Representative composite color maps using the MBR as measured with LSFG with a 200-pixel diameter encircling the optic nerve head. (**B**) Enlargement of the optic nerve head area. The red color indicates high MBR, and the blue color indicates low MBR. (**C**) LSFG Analyzer Software automated binarization of the image into vessels (white area) and tissue (black area). (**D**) Fluctuations of MBR throughout a 118-frame scan with a calculation of waveform parameters: beat strength (BS), beat strength over MBR (BOM), Max MBR, Min MBR, and Max–Min MBR. (**E**) The total retinal artery and vein analyzer (TRAVA) automated the calculation of the total retinal flow index divided by the vessel width (TRFI/w) for the top four highest velocity arteries only (top four in bright red).

**Figure 2 cells-12-02685-f002:**
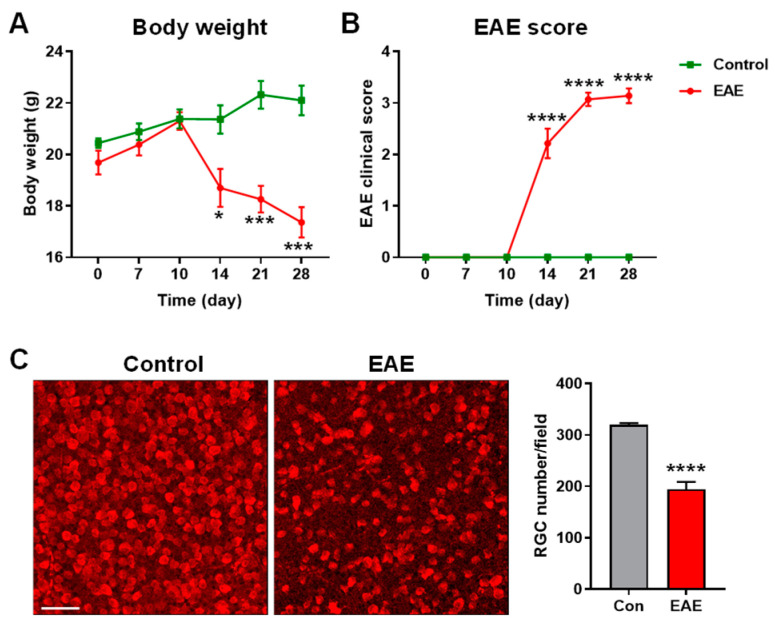
Evaluation of the EAE mice model. The EAE model was induced in WT mice. Body weight (**A**) and EAE clinical score (**B**) were measured at 0, 7, 10, 14, 21, and 28 dpi. *n* = 5–7. (**C**) Representative images of RBPMS-positive RGCs from the control and EAE mice at 28 dpi. The bar graph represents the quantification of the RGC number. Scale bar = 50 µm. *n* = 9–14; * *p* < 0.05, *** *p* < 0.001, **** *p* < 0.0001.

**Figure 3 cells-12-02685-f003:**
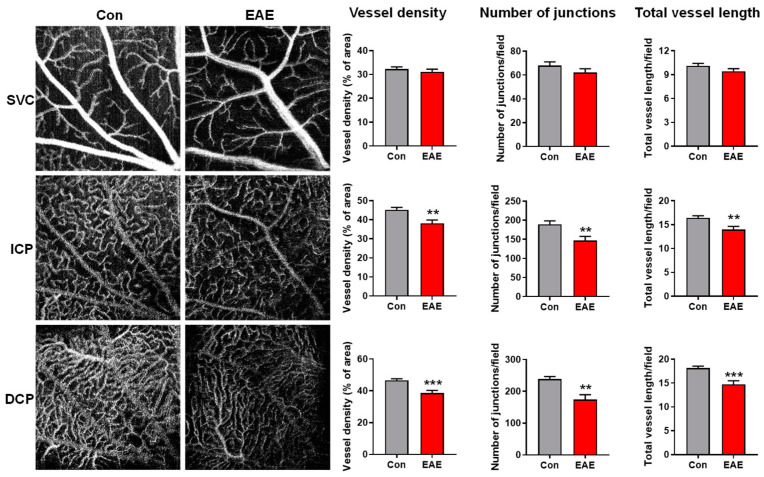
EAE impairs retinal vascular function. The EAE model was induced in WT mice. Retinal vasculature was examined via OCTA at 14 dpi, and representative OCTA images were shown. Vessel density, vessel branch points (number of junctions), and total vessel length were quantified with AngioTool. *n* = 14–16; ** *p* < 0.01, *** *p* < 0.001. SVC: superficial vascular complex; ICP: intermediate capillary plexus; DCP: deep capillary plexus.

**Figure 4 cells-12-02685-f004:**
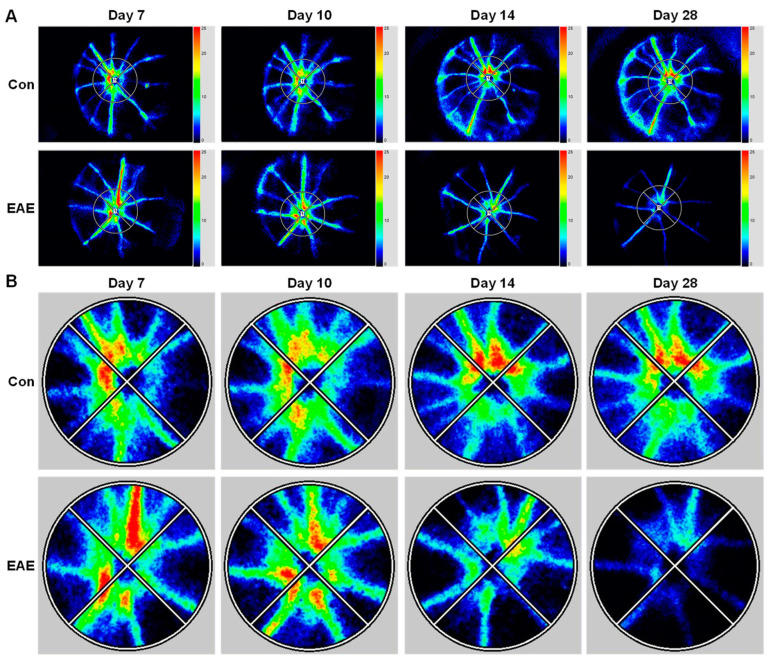
EAE impairs retinal blood flow. The EAE model was induced in WT mice. Retinal blood flow was examined with LSFG at 7, 10, 14, and 28 dpi. (**A**) Representative composite color maps using the MBR via LSFG with a 200-pixel diameter encircling the optic nerve head. (**B**) Enlargement of the optic nerve head area. The red color indicates high MBR, and the blue color indicates low MBR.

**Figure 5 cells-12-02685-f005:**
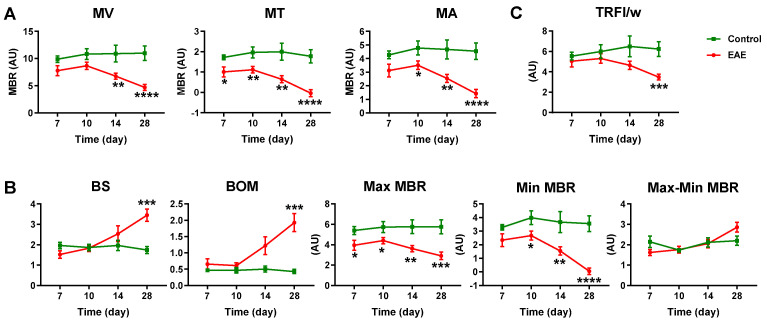
EAE decreases retinal blood flow velocity and increases vascular resistance. LSFG images were analyzed using the LSFG Analyzer software (version 3.00) for the MBR in MV, MT, and MA (**A**), waveform parameters BS, BOM, Max MBR, Min MBR, and Max–Min MBR in the overall area of the ONH (**B**); and TRFI/w (**C**). *n* = 9–16; * *p* < 0.05, ** *p* < 0.01, *** *p* < 0.001, **** *p* < 0.0001. MV: MBR in the vessel area; MT: MBR in the tissue area; MA: MBR in all areas.

**Figure 6 cells-12-02685-f006:**
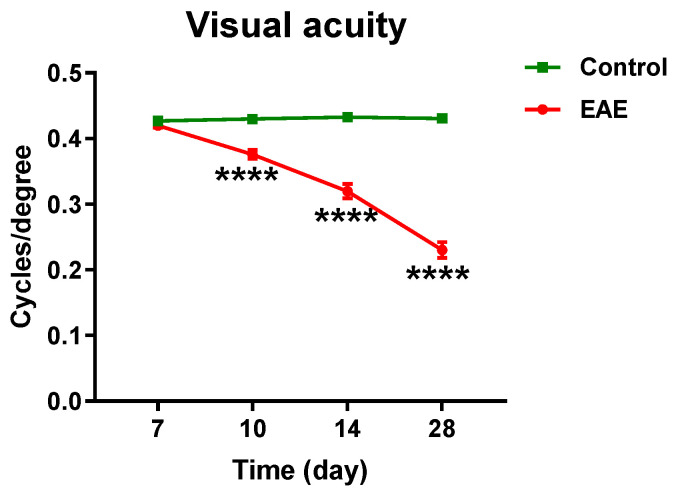
Visual acuity in the EAE mice. The EAE model was induced in WT mice. Visual acuity was evaluated at 7, 10, 14, and 28 dpi. *n* = 9–14; **** *p* < 0.0001.

**Figure 7 cells-12-02685-f007:**
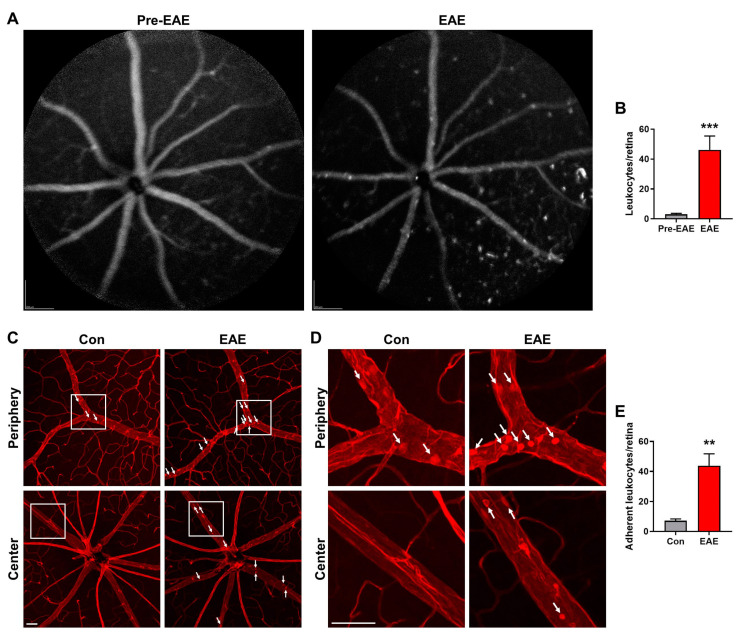
Leukocyte attachment is increased in the retina of the EAE mice. (**A**,**B**) SLO imaging was performed right before and 7 days after EAE induction, and stationary leukocytes in the retinas were quantified. (**C**–**E**) Leukostasis was performed 14 days after EAE induction. Representative images of leukostasis in peripheral and central retinas from the control and EAE mice were shown. Images are enlarged from the square of C. White arrows indicate adherent leukocytes. The bar graph represents the number of adherent leukocytes per retina. Scale bar = 50 µm. *n* = 6; ** *p* < 0.01, *** *p* < 0.001.

**Table 1 cells-12-02685-t001:** LSFG parameters with their associated descriptions.

Parameter	Description
MV	MV indicates intravascular flow velocity for large retinal vessels (i.e., arteries, veins, arterioles, venules) determined by an automated MBR threshold.
MT	MT indicates blood flow velocity within tissue (i.e., capillaries) as predefined by the automated MBR threshold.
MA	MA indicates blood flow velocity in both vascular and tissue areas.
BS	BS indicates fluctuation of blood flow at the onset of a cardiac cycle and is proportional to the peripheral vascular resistance.
BOM	The BOM is defined as the BS divided by the average MBR value in a rubber band. This parameter expresses peripheral vascular resistance.
Max MBR	The maximum MBR obtained during the 4 s (118 frames) scan.
Min MBR	The minimum MBR obtained during the 4 s (118 frames) scan.
Max–Min MBR	The minimum MBR obtained is subtracted from the maximum MBR to obtain the Max–Min MBR. An elevated Max–Min MBR suggests large variation in blood flow.
TRFI/w	The volume of blood flow occurring in four vessels of the retina with the highest blood flow.

## Data Availability

All data are presented in the main manuscript and the manuscript figures.

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
