# Peer review of "Non-Invasive Evaluation of Retinal Vascular Alterations in a Mouse Model of Optic Neuritis Using Laser Speckle Flowgraphy and Optical Coherence Tomography Angiography"

_cells, 2023, doi:10.3390/cells12232685_

Round 1

Reviewer 1 Report

Comments and Suggestions for Authors

The article entitled “Non-invasive Evaluation of Retinal Vascular Alterations in a Mouse Model of Optic Neuritis by Laser Speckle Flowgraphy and Optical Coherence Tomography Angiography” for the first time demonstrates the usefulness of LSFG to detect impaired retinal circulation associated with optic neuritis. The results are very promising and will be sure to promote LSFG measurement in monitoring patients suffering from multiple sclerosis. However, there are still minor issues mentioned below.

The evaluation of leukostasis was performed by using the eyeballs enucleated at 14 dpi. If leukostasis is a cause of impaired retinal circulation in this model, the abnormalities should be detected at earlier time points such as 7 dpi. Have you done this evaluation at any earlier time points than 14 dpi?

In Discussion, LSFG and OCTA is by far less invasive, less expensive and less time-consuming compared to MRI and CSF analysis. I totally agree with it. However, MRI and CSF analysis are inevitable to precisely diagnose optic neuritis associated with MS. Since optic neuropathies are caused by various disorders not only MS but trauma, sinusitis and so on. This might be a limitation in using LSFG and OCTA.

In materials and methods, the total number of animals used in this study should be written in the section 2.1 Animals.

Author Response

1. The evaluation of leukostasis was performed by using the eyeballs enucleated at 14 dpi. If leukostasis is a cause of impaired retinal circulation in this model, the abnormalities should be detected at earlier time points such as 7 dpi. Have you done this evaluation at any earlier time points than 14 dpi?

RESPONSE: We have added the evaluation of leukocyte attachment at 7 dpi (new Figure 7A-7B) (ages 9-10). The marked increase in leukocyte attachment is consistent with previous report that in the optic nerve an upregulation of inflammatory cytokines occurs as early as 7 dpi (Liu et al. Neuromolecular Med. 2015 Dec;17(4):391-403. PMID: 26318182). We have added this source (lines 333-335).

2. In Discussion, LSFG and OCTA is by far less invasive, less expensive and less time-consuming compared to MRI and CSF analysis. I totally agree with it. However, MRI and CSF analysis are inevitable to precisely diagnose optic neuritis associated with MS. Since optic neuropathies are caused by various disorders not only MS but trauma, sinusitis and so on. This might be a limitation in using LSFG and OCTA.”

RESPONSE: We agree with the reviewer’s concern and have discussed the limitation of using LSFG and OCTA in the Discussion (lines 437-441).

3. In materials and methods, the total number of animals used in this study should be written in the section 2.1 Animals.”

RESPONSE: This information has been added to the Materials and Methods as suggested.

Reviewer 2 Report

Comments and Suggestions for Authors

This manuscript describes a non-invasive evaluation of retinal vascular changes in EAE mice using a combination of laser speckle flowgraphy and optical coherence tomography angiography. Overall the manuscript is written nicely with clear descriptions of the background, methods, and results. However, the novelty and impact of the study seems not significant. The study validated and characterized the vascular alterations in the retina of the EAE mice, which is consistent with the clinical observations. The validation itself is novel but does not bring any new insight on the mechanism or therapy for this disease. The authors suggested a potential to develop the image-based biomarkers for the diagnosis and therapeutic monitoring of the disease, which is very interesting but lacks supporting data showing the association between the leukostasis/RGC injury/EAE progression and the image parameters.

Comments on the Quality of English Language

The quality of English writing is good. 

Author Response

1. This manuscript describes a non-invasive evaluation of retinal vascular changes in EAE mice using a combination of laser speckle flowgraphy and optical coherence tomography angiography. Overall the manuscript is written nicely with clear descriptions of the background, methods, and results. However, the novelty and impact of the study seems not significant. The study validated and characterized the vascular alterations in the retina of the EAE mice, which is consistent with the clinical observations. The validation itself is novel but does not bring any new insight on the mechanism or therapy for this disease. The authors suggested a potential to develop the image-based biomarkers for the diagnosis and therapeutic monitoring of the disease, which is very interesting but lacks supporting data showing the association between the leukostasis/RGC injury/EAE progression and the image parameters.”

RESPONSE: While clinical studies suggest that retinal vessel density and blood perfusion are altered in ON, the data is often contradictory. Some studies indicated that eyes from MS patients had significantly decreased retinal vessel density in the superficial vascular complex and deep vascular complex compared with healthy controls, and that vessel density reduction was even greater in the eyes with ON, associated with reduced oxygen extraction and retinal blood flow. However, other studies did not find any vessel density reduction in MS eyes compared with healthy eyes or a difference between MS eyes with vs without ON. There are also many inconsistencies regarding the location of and mechanisms underlying these vascular changes. Considering clinical studies cannot control confounding variables such as age, diet, environment, disease initiation, duration and modifying treatments, it is not surprising that conflicting results were reported.

Compared with clinical studies, animal models allow for controlling of the aforementioned confounding variables, making it easier to investigate pathological changes and their underlying mechanisms in diseases. Our study provides the first evidence that the retina undergoes significant hypoperfusion during the course of optic neuritis in a mouse EAE model. This study is significant because it clears the contradictions in the clinical studies and supports clinical studies which demonstrate that decreased retinal blood flow velocity by retinal function imager and decreased retinal vascular density in OCTA scans are present in patients with optic neuritis. Moreover, our study is novel in that it is the first to demonstrate noninvasive imaging modalities can detect optic neuritis prior to the onset of impairment in visual acuity or progression of systemic signs of EAE. Indeed, visual acuity and EAE score did not significantly decline until 10 and 14 dpi in EAE mice (Figure 6, Figure 2); however, noninvasive imaging can detect vascular changes in EAE mice as early as 7 dpi (Figure 4, 5). This offers a unique opportunity for initiation of asymptomatic testing for patients at high risk for optic neuritis, early initiation of treatment, and potentially superior patient outcomes. The discussion has been amended to emphasize the importance and innovation of this study (lines 444-451).

Although our study suggested a potential to develop the image-based biomarkers for the diagnosis and therapeutic monitoring of the disease and to further explore retinal vascular alterations in an EAE model to unravel its underlying mechanisms, it would take many years and enormous efforts and resources to conduct a lot of comprehensive and explorative studies to accomplish these goals. These studies are beyond the scope of the current manuscript. However, we are continuing working on these directions and will publish the findings in the future manuscripts.  

2. Minor editing of English language is required.”

RESPONSE: We have made several edits and improved the grammar of our manuscript.

Round 2

Reviewer 2 Report

Comments and Suggestions for Authors

The authors' response and modifications clarify the significance of the study. This manuscript is ready for publication.